

# Gut microbiota diversity and composition in children with autism spectrum disorder: associations with symptom severity

Qinghuang Zeng[1,2,*], Yisheng Hu[3,4,*], Leiying Xie[3], Xinyi Zhang[3], Yun Huang[3], Jianbin Ye[3,4], Shouan Wang[3] and Jia Xu[3,5]

[1] School of Clinical Medicine, Fujian Medical University, Fuzhou, China
[2] Affiliated Hospital of Putian University, Putian, China
[3] School of Basic Medicine Science, Key Laboratory of Translational Tumor Medicine in Fujian Province, Putian University, Putian, China
[4] School of Pharmacy, Putian University, Putian, China
[5] School of Basic Medicine Science, Fujian Medical University, Putian, China
[*] These authors contributed equally to this work.

## ABSTRACT

**Background.** Autism spectrum disorder (ASD) is a prevalent neurodevelopmental disorder impairing social and communication skills. Gut microbiota has become key in understanding ASD pathophysiology. However, the relationship between the ASD symptoms and alternation of gut microbiota still remains unknow. We hypothesize that the composition of gut microbiota in children with ASD may be strongly associated with the severity of their symptoms.

**Methods.** Here, fecal samples from children (divided in to three groups: neurotypical, severe ASD and mild ASD) at a hospital were collected. The symptoms of ASD were assessed by an experienced pediatric neurologist, and the severity of the symptoms in children with ASD was determined based on the assessment scores. Then the diversity and composition of gut microbiota were detected by high-throughput sequencing.

**Results.** In total, 2,021 amplicon sequence variants (ASVs) were obtained from 46 fecal samples, with highest in the neurotypical group. Alpha diversity in bacteria differed between severe and mild ASD. Microbiota health and dysbiosis indices varied with ASD severity. Beta diversity indicated that severe ASD differed from others, and mild ASD was closer to neurotypical in community structure. At the phylum level, Firmicutes was the dominant bacteria but abundances differed in different groups, and Ascomycota increased in severe ASD fungi. At the genus level, groups had distinct dominants, and mild ASD microbiota resembled that of neurotypical children. Function prediction revealed differences in bacteria and fungi, with severe ASD having higher amino acid metabolism, lower cofactor/vitamin metabolism, and more undefined saprotrophs.

**Conclusion.** This study revealed gut microbiota differences between ASD children (varying symptoms) and neurotypical ones, showing milder ASD closer in microbiota aspects. It offers insights for exploring ASD pathogenesis and devising interventions.

Corresponding authors
Jianbin Ye, happye1986@163.com
Shouan Wang, wangshouanjms@126.com

## INTRODUCTION

Autism spectrum disorder (ASD), a highly intricate neurodevelopmental condition, is marked by deficits in social interaction and communication skills, along with the manifestation of restricted, repetitive behaviors and narrow interests (*Fattorusso et al., 2019*; *Sharon et al., 2019*). Typically diagnosed during early childhood years, ASD can exert a profound and enduring influence on an individual's capacity to lead an independent life and form meaningful social connections (*Liu et al., 2019*). Over the past several decades, there has been a persistent upward trend in the prevalence of ASD. As reported by the Centers for Disease Control and Prevention (CDC) in 2020, roughly 1 out of every 36 children aged 8 had been identified as having ASD, which has emerged as a major public health issue (*Maenner et al., 2020*). This escalating prevalence not only imposes a significant burden on those affected and their families but also presents formidable challenges to educational institutions, healthcare providers, and social support frameworks. Children with ASD invariably necessitate specialized educational curricula, behavioral interventions, and holistic medical attention to cater to their specific requirements (*Al-Beltagi, 2024*; *Weiss, Wingsiong & Lunsky, 2014*).

The gut microbiota, constituting a complex assemblage of microorganisms inhabiting the gastrointestinal tract, has surfaced as a pivotal domain of research in unraveling the pathophysiology underlying ASD (*Ristori et al., 2019*; *Su et al., 2024*; *Zhang et al., 2015*). In recent times, investigations have furnished compelling proof of variances in the composition of the gut microbiota between children with ASD and their neurotypical counterparts. To illustrate, *Strati et al. (2017)* disclosed significant disparities in the relative abundances of diverse bacterial taxa. Notably, there was a decline in the Firmicutes/Bacteroidetes ratio, accompanied by alterations in the abundances of particular genera such as *Alistipes*, *Bilophila*, and *Dialister* (*Strati et al., 2017*). Study showed that children with ASD manifested lower bacterial diversity and richness, in addition to discrepancies in the abundances of certain families and genera (*Ma et al., 2019*). These revelations have spurred the exploration of prospective treatment modalities centered around the gut microbiota. Probiotic supplementation, designed to introduce beneficial bacteria into the gut environment, has evinced some potential in alleviating gastrointestinal symptoms and potentially modulating behavioral patterns in ASD patients (*He et al., 2023*; *Wang et al., 2020*). *Shaaban et al. (2018)* undertook a prospective, open-label study and documented favorable outcomes of probiotics in children with ASD. Moreover, fecal microbiota transplantation (FMT), entailing the transfer of fecal matter from a healthy donor to an ASD patient, has also been probed as a conceivable therapeutic avenue. *Kang et al. (2017)* reported that microbiota transfer therapy led to improvements in both gastrointestinal and autism symptoms within the framework of an open-label study.

Several bacterial and fungal taxa have emerged as prospective biomarkers linked to ASD. At the bacterial level, beyond the previously mentioned modifications in Firmicutes and Bacteroidetes, consistent reports have detailed changes in the abundances of *Clostridium* species. Research efforts have discerned differences in the fecal microflora of autistic children as opposed to those of controls, with particular shifts in *Clostridium* species being

noted (*Finegold et al., 2002*; *Parracho et al., 2005*). Additionally, certain investigations have proposed a role for other taxa; for instance, *Akkermansia muciniphila*, which is involved in gut barrier function and immune regulation. Notably, relatively low abundances of *Akkermansia muciniphila* have been observed in the feces of children with ASD (*Zou et al., 2020*). In the realm of the mycobiome, studies discovered an augmentation in the fungal Ascomycota phylum and *Candida albicans* in ASD patients when compared to their non-autistic siblings (*Liu et al., 2019*; *Retuerto et al., 2024*). Furthermore, alterations in the levels of *Saccharomyces* and other fungal genera have also been documented in some studies, albeit with less consistency in the results (*Li et al., 2024*; *Zou et al., 2021*).

Investigating the gut microbiota in children with ASD holds paramount significance as it paves the way for deciphering the underlying mechanisms of this disorder and devising efficacious treatment strategies. Numerous studies have demonstrated that gut microbiota composition affects human neurotransmission and influences the development and progression of autism spectrum disorder symptoms (*Bamicha, Pergantis & Drigas, 2024*). The gut-brain axis (GBA), a two-way communication network interconnecting the gut and the brain, is widely acknowledged to play a pivotal role in the pathophysiology of ASD as well as other diseases related to the central nervous system (*Sharon et al., 2016*; *Wang et al., 2023*). For example, relevant review articles suggest that gut dysbiosis in ASD contributes to increased intestinal permeability ("leaky gut") and is associated with neurodevelopmental impairments, where bacterial metabolites may disrupt brain function *via* the gut-brain axis (*Fowlie, Cohen & Ming, 2018*). Alterations within the gut microbiota possess the potential to impact this axis, subsequently triggering changes in brain function and behavior (*Wang, Yang & Liu, 2023*). For example, the generation of short-chain fatty acids (SCFAs), including acetate, propionate, and butyrate, by gut bacteria can exert an influence on neurotransmitter synthesis and neural development. Numerous studies have detected modified levels of SCFAs in children with ASD, which might contribute to the manifestation of neurological symptoms in these patients (*Lagod & Naser, 2023*; *Liu et al., 2019*). Another study using multi-omics analysis revealed the functional architecture of the GBA in ASD, identifying specific microbial were closely linked to pro-inflammatory cytokines and restrictive dietary patterns, and the ASD-associated signals observed in age- and sex-matched cohorts disappeared in sibling-matched cohorts, highlighting the synergistic mechanisms of the microbiome-metabolite-immune-neural pathways in ASD (*Morton et al., 2023*). Moreover, the gut microbiota is capable of modulating the immune system, and immune dysregulation has been associated with ASD (*De Angelis et al., 2015*; *Fattorusso et al., 2019*). By grasping the specific modifications in the gut microbiota correlated with ASD, it becomes feasible to pinpoint early biomarkers for the disorder, thereby facilitating earlier diagnosis and intervention (*Chen et al., 2022*). This, in turn, could potentially culminate in more effective treatments and enhanced long-term prognoses for children afflicted with ASD.

Thus far, microbiome-targeted therapies for ASD (such as probiotics and fecal microbiota transplantation/FMT) show therapeutic potential, they still require rigorous clinical validation (*Fowlie, Cohen & Ming, 2018*). In other side, extensive researches on the association between ASD and gut microbiota has demonstrated that microbial alterations

may influence neurodevelopment through the GBA. However, these studies still lack sufficient evidence to establish a positive correlation between the degree of gut microbial changes and ASD severity. Notably, few studies have systematically compared differences among severe ASD, mild ASD, and neurotypical children—a comparison that could provide valuable insights for clinical diagnosis and treatment of ASD.

This study analyzed the gut microbiota of children with different ASD symptoms and neurotypical children. We aim to reveal the relationship between gut microbiota and ASD symptoms, which will supply beneficial data and support for the relevant treatments of ASD symptoms.

## MATERIALS AND METHODS

### Samples collection

Approximately 60 samples were collected from children who had not undergone antibiotic treatment at the Affiliated Hospital of Putian University. For all the ASD patients involved in this study, their samples were gathered during outpatient consultations. The control group samples were sourced from neurotypical children who had not received any medications within one month prior to sample collection. Fresh fecal samples were obtained through rectal swabs. Subsequently, certain fecal samples from ASD patients and neurotypical children were excluded as a result of insufficient sample weight or suboptimal sample quality following DNA amplification.

Consequently, only 46 fecal samples were utilized for further sequencing and analysis, with 16 originating from neurotypical children, 15 from children with moderate/severe autism, and 15 from children with mild autism. The autistic children were admitted to the Affiliated Hospital of Putian University. Subsequently, the diagnosis of ASDs was established in accordance with the Diagnostic and Statistical Manual of Mental Disorders, 5th Edition (*First, 2013*). Additionally, evaluations were conducted using the Autism Diagnostic Observation Schedule and the Autism Behaviour Checklist. The scores of the Childhood Autism Rating Scale (CARS) (*Schopler et al., 1980*) were calculated by an experienced child neuropsychiatrist. The samples were stored in five ml tubes and promptly frozen at $-80\ ^{\circ}C$ until required for use. This study received approval from the Ethics Committee of the Affiliated Hospital of Putian University (Approval No.: 2023090-1), and the informed consent forms were signed by the parents. The detailed information and scoring criteria of all samples are presented in Table 1.

### Main reagents and instruments

This study employed the following reagents and instruments: The FastPure Stool DNA Isolation Kit, provided by Shanghai Majorbio Bio-Pharm Technology Co., Ltd (Shanghai, China). The FastPfu Polymerase, sourced from Beijing TransGen Biotech Co., Ltd (Beijing, China). The NEXTFLEX Rapid DNA-Seq Kit, manufactured by Bioo Scientific (Austin, TX, USA). The T100 Thermal Cycler, produced by Bio-Rad Laboratories, Inc. (Hercules, CA, USA). The JY600C Double Stable Time Electrophoresis Apparatus, supplied by Beijing Junyi Dongfang Electrophoresis Equipment Co., Ltd (Beijing, China). The Synergy HTX, made by Biotek (Winooski, VT, USA). The Illumina's NextSeq 2000 PE300 platform, which

**Table 1 The basic information of all samples.**

| Group | Sample\ID | ASD or neurotypical | Age (years) | Gender | Cars score |
|---|---|---|---|---|---|
| G1 | H_2 | Neurotypical | 8 | Male | Neurotypical children |
| | H_3 | Neurotypical | 6 | Female | Neurotypical children |
| | H_4 | Neurotypical | 8.5 | Female | Neurotypical children |
| | H_7 | Neurotypical | 10 | Male | Neurotypical children |
| | H_9 | Neurotypical | 6 | Female | Neurotypical children |
| | H_11 | Neurotypical | 7 | Female | Neurotypical children |
| | H_13 | Neurotypical | 5.5 | Male | Neurotypical children |
| | H_15 | Neurotypical | 6.5 | Female | Neurotypical children |
| | H_18 | Neurotypical | 7 | Male | Neurotypical children |
| | H_19 | Neurotypical | 8 | Female | Neurotypical children |
| | H_21 | Neurotypical | 5 | Male | Neurotypical children |
| | H_23 | Neurotypical | 7 | Female | Neurotypical children |
| | H_24 | Neurotypical | 6 | Male | Neurotypical children |
| | H_25 | Neurotypical | 7 | Female | Neurotypical children |
| | H_26 | Neurotypical | 5 | Male | Neurotypical children |
| | H_28 | Neurotypical | 6 | Female | Neurotypical children |
| G2 | C_2 | Severe ASD | 6 | Female | 45 |
| | C_4 | Severe ASD | 5.5 | Male | 40 |
| | C_5 | Severe ASD | 4.5 | Male | 43 |
| | C_6 | Severe ASD | 6 | Male | 41 |
| | C_8 | Severe ASD | 7 | Female | 38 |
| | C_26 | Severe ASD | 6 | Female | 51 |
| | C_27 | Severe ASD | 8 | Male | 53 |
| | C_28 | Severe ASD | 6.5 | Female | 54 |
| | C_29 | Severe ASD | 7.5 | Male | 48 |
| | C_30 | Severe ASD | 8 | Male | 56 |
| | C_31 | Severe ASD | 9 | Female | 44 |
| | C_33 | Severe ASD | 11 | Female | 55 |
| | C_35 | Severe ASD | 7 | Female | 46 |
| | C_36 | Severe ASD | 7 | Male | 50 |
| | C_37 | Severe ASD | 6 | Male | 50 |
| G3 | C_3 | Mild ASD | 7 | Male | 30 |
| | C_11 | Mild ASD | 8 | Female | 30 |
| | C_13 | Mild ASD | 5.5 | Female | 32 |
| | C_14 | Mild ASD | 6.5 | Male | 35 |
| | C_15 | Mild ASD | 8 | Male | 34 |
| | C_16 | Mild ASD | 7 | Male | 31 |
| | C_17 | Mild ASD | 6.5 | Female | 30 |
| | C_19 | Mild ASD | 8 | Male | 32 |
| | C_20 | Mild ASD | 9 | Female | 30 |

**Table 1** (*continued*)

| Group | Sample\ID | ASD or neurotypical | Age (years) | Gender | Cars score |
|-------|-----------|---------------------|-------------|--------|------------|
| | C_21 | Mild ASD | 6 | Male | 32 |
| | C_22 | Mild ASD | 7.5 | Male | 33 |
| | C_23 | Mild ASD | 8 | Female | 32 |
| | C_24 | Mild ASD | 6.5 | Female | 35 |
| | C_25 | Mild ASD | 5 | Male | 35 |
| | C_32 | Mild ASD | 7 | Female | 34 |

was utilized for high-throughput sequencing and the sequencing work was completed by Shanghai Majorbio Bio-Pharm Technology Co., Ltd (Shanghai, China).

## Sample DNA extraction

The total genomic DNA of the microbial community present in the fecal samples of children with ASD (designated as the disease group) and those of neurotypical children (termed the neurotypical group) was extracted in strict accordance with the operating instructions provided by the FastPure Stool DNA Isolation Kit (Vazyme, Nanjing, China). Subsequently, the integrity of the extracted genomic DNA was examined through 1% agarose gel electrophoresis. Meanwhile, the DNA concentration and purity were quantified using the NanoDrop2000 instrument (manufactured by Thermo Fisher Scientific, Waltham, MA, USA).

## PCR amplification and sequencing library construction

Taking the DNA extracted from the fecal samples as described above as a template, the V3–V4 variable region of the 16S rRNA gene was amplified through polymerase chain reaction (PCR). The upstream primer 338F (5′-CTCCTACGGGAGGCAGCAG-3′) and the downstream primer 806R (5′-GGACTACHVGGGTWTCTAAT-3′), which carried the Barcode sequence, were utilized for this amplification. For amplification of the ITS gene, the upstream primer ITS3F (5′-GCATCGATGAAGAACGCAGC-3′) and the downstream primer ITS4R (5′-TCCTCCGCTTATTGATATGC-3′) were used. The PCR amplification program was set as follows: an initial pre-denaturation step at 95 °C for 3 min, followed by 27 cycles. Each cycle comprised denaturation at 95 °C for 30 s, annealing at 55 °C for 30 s, and extension at 72 °C for 30 s. Subsequently, a stable extension was carried out at 72 °C for 10 min, and finally, the samples were stored at 4 °C. The PCR instrument used in this process was the T100 Thermal Cycler.

The PCR reaction system using the TransStart FastPfu DNA polymerase purchased from TransGen (2* TransStart FastPfu PC, AS221-02, Beijing, China) and carried out according to the manufacture's instruction, which includes following components: DNA polymerase (0.4 μL), template DNA (10 ng), the upstream and downstream primer were both 0.8 μL (5 μM), 2.5 mM dNTPs (2 μL), 5XTransStart FastPfu buffer (4 μL), and then adjust the total volume to 20 μL using the proper solvent. Each sample was replicated three times to ensure reproducibility. The PCR products of the same sample were combined and then recovered by means of 2% agarose gel electrophoresis. Subsequently, the recovered
products were purified. The size of the resulting band fragments was determined by 2% agarose gel electrophoresis, and the recovered products were detected and quantified using the Synergy HTX device. The purified PCR products were then utilized to construct a sequencing library with the help of the NEXTFLEX Rapid DNA-Seq Kit. This construction process involved several key steps: first, linker ligation was carried out; then, magnetic beads were employed to screen and eliminate linker self-ligation fragments; next, the library template was enriched through PCR amplification; and finally, the PCR products were recovered using magnetic beads to obtain the final, ready-to-use library. Sequencing was performed on the Illumina's NextSeq 2000 PE300 platform (undertaken by Shanghai Majorbio Bio-Pharm Technology Co., Ltd, Shanghai, China). After sequencing, the original data was uploaded to the NCBI SRA database for further analysis and sharing within the scientific community (PRJNA1207206) and the data was also deposited in the FigShare under the DOI: https://doi.org/10.6084/m9.figshare.28303508.

## High-throughput sequencing data analysis

The paired-end raw sequencing reads underwent quality control procedures using fastp (available at https://github.com/OpenGene/fastp), and were subsequently assembled with FLASH (accessible at http://www.cbcb.umd.edu/software/flash, version 1.2.11). The detailed operations were carried out as follows: bases at the read termini with a quality value lower than 20 were filtered out. A window of 50 base pairs (bp) was set, and if the average quality value within this window fell below 20, bases were truncated from the end of the window. Reads with a length less than 50 bp after quality control were filtered, and those containing N bases were removed. Based on the overlap relationship between the paired-end reads, paired reads were merged into a single sequence, with a minimum overlap length of 10 bp. The maximum allowable mismatch ratio within the overlap region of the assembled sequence was set at 0.2, and sequences not meeting this criterion were screened out. Samples were distinguished according to the barcode and primers at both ends of the sequence, and the sequence direction was adjusted. The permitted number of barcode mismatches was 0, while the maximum number of primer mismatches was 2.

The optimized sequences underwent quality control and assembly before being processed with the DADA2 plugin for denoising (*Callahan et al., 2016*), which resulting in ASVs. To minimize bias from uneven sequencing depth in downstream alpha and beta diversity analyses, all samples were rarefied to 26,359 reads per sample. This normalization achieved an average Good's coverage of 99.09%, indicating robust representation of microbial diversity. Based on the Silva 16S rRNA gene database (version 138) and unite9.0/its_fungi, the ASVs were taxonomically classified using the naive Bayes in Qiime2 (*Bolyen et al., 2019*). 16S rRNA function prediction analysis was conducted using PICRUSt2 (version 2.2.0), and the Funguild (version 1.0) was employed to evaluate the differential fungal microecology among different groups.

## Statistical analysis

Data processing and statistical analyses were conducted using the Majorbio Cloud Platform (https://cloud.majorbio.com). Alpha diversity metrics, such as the Shannon and Ace indices,

were calculated using mothur (http://www.mothur.org/wiki/Calculators, v1.30.2) (*Schloss et al., 2009*). Differences in alpha diversity across groups were evaluated using the Wilcoxon rank-sum test. To examine beta diversity, microbial community dissimilarities were visualized *via* principal coordinate analysis (PCoA) and non-metric multidimensional scaling (NMDS), both based on Bray-Curtis distances. Additionally, PERMANOVA was applied to determine the statistical significance of structural differences between sample groups.

## RESULTS

### Differences in gut bacterial and fungal diversity between ASD and neurotypical children

In total, 46 fecal specimens were successfully sequenced. Following the sequence quality control procedures, 1,713,362 sequences were acquired from all samples (Table S1). A combined total of 2,021 amplicon sequence variants (ASV) were obtained across all samples, with 715 ASV originating from the neurotypical group, 666 ASV from the severe ASD group, and 640 ASV from the mild to moderate ASD group. Notably, a higher number of ASV were obtained in the neurotypical group compared to the ASD patients. The rarefaction curve demonstrated that the sequencing depth was sufficient as it plateaued for all samples (Fig. S1), signifying that the majority of microbiota species had been captured.

Alpha diversity was computed to appraise the microbial diversity within the samples. The Kruskal–Wallis H test methodology was employed to calculate and comparatively analyze the inter-group differences in Alpha diversity of both bacteria and fungi among the distinct groups. In the 16S analysis, it was revealed that there were pronounced differences in bacterial diversity among the different groups. Specifically, the Ace index of children with severe ASD did not exhibit a significant difference from that of the neurotypical group ($P > 0.05$), yet it did display a significant difference from the Ace index of children with mild ASD ($P < 0.05$) (Fig. 1). Likewise, in the ITS analysis, no significant differences were observed among the three groups ($P < 0.05$).

However, upon analyzing the gut microbiota health index (Fig. 2), it was uncovered that significant differences existed in the microbiota health index among different groups, and this disparity bore no significant correlation with diversity. When comparing group G1 (the neurotypical group) to group G2 (the group of children with severe ASD), it was observed that the majority of samples in group G1 exhibited a higher gut microbiota health index (GMHI). Likewise, group G3 (the group of children with mild ASD) had a comparatively higher GMHI than group G2. Additionally, we conducted a further analysis of the differences in the microbiota dysbiosis index among the three groups of samples. It was found that, regardless of whether considering bacteria or fungi, there were pronounced differences between group G1 and the other two groups of children with ASD. Nevertheless, this difference appeared to be diminishing as the microbiota dysbiosis index (MDI) of group G1 and group G3 was closer. These results signify that the gut microbiota health index of children with ASD has undergone substantial alterations. In particular, the degree of microbiota dysbiosis is closely associated with the severity of ASD symptoms.

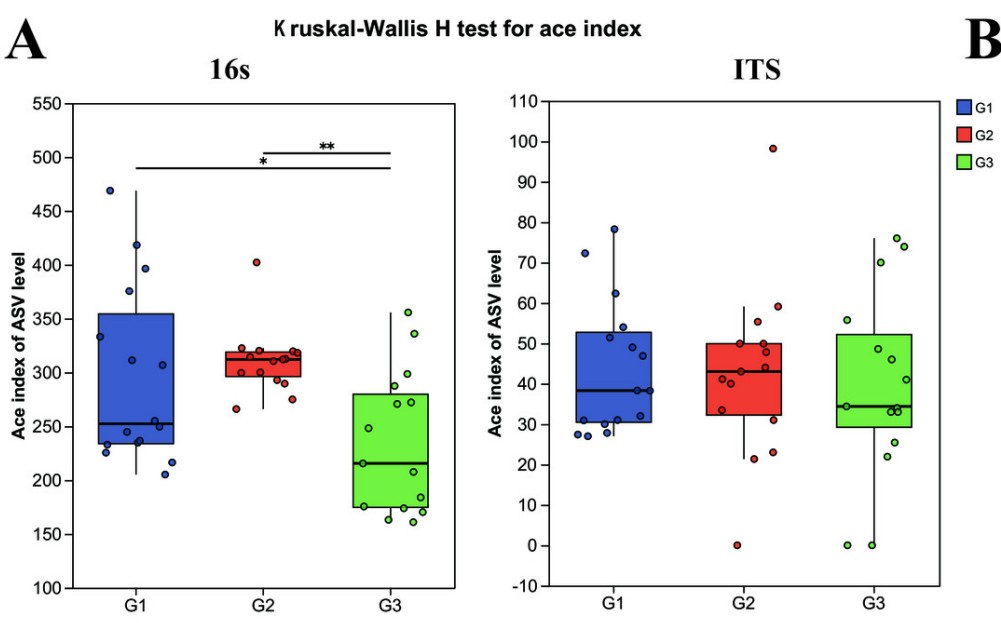

**Figure 1 The alpha diversity indices of bacteria (16s) and fungi (ITS) in different groups.** G1 represents neurotypical children; G2 represents severe ASD children; G3 represents mild ASD children.

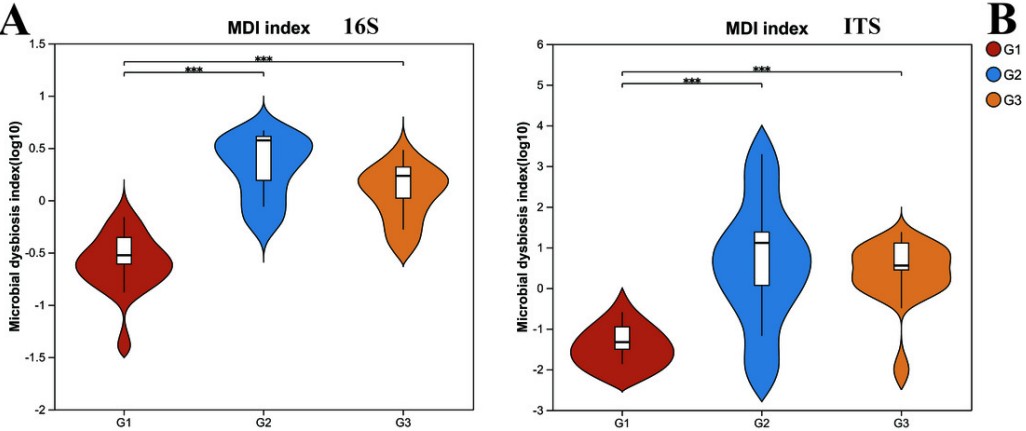

**Figure 2 The microbiota dysbiosis index (MDI) of bacteria (16s) and fungi (ITS) in different groups.** G1 represents neurotypical children; G2 represents severe ASD children; G3 represents mild ASD children.

Beta diversity was utilized to assess the diversity of microbial community structure among different groups. As depicted in Fig. 3, the results of the 16S analysis indicate that there are significant differences in bacterial beta diversity between group G2 and both group G1 and group G3 ($P < 0.05$), whereas there is no significant difference between group G1 and group G3. Similarly, the ITS analysis reveals that there are significant differences in fungal beta diversity between group G2 and both group G1 and group G3 ($P < 0.05$), yet the difference between group G1 and group G3 becomes less pronounced. Such findings

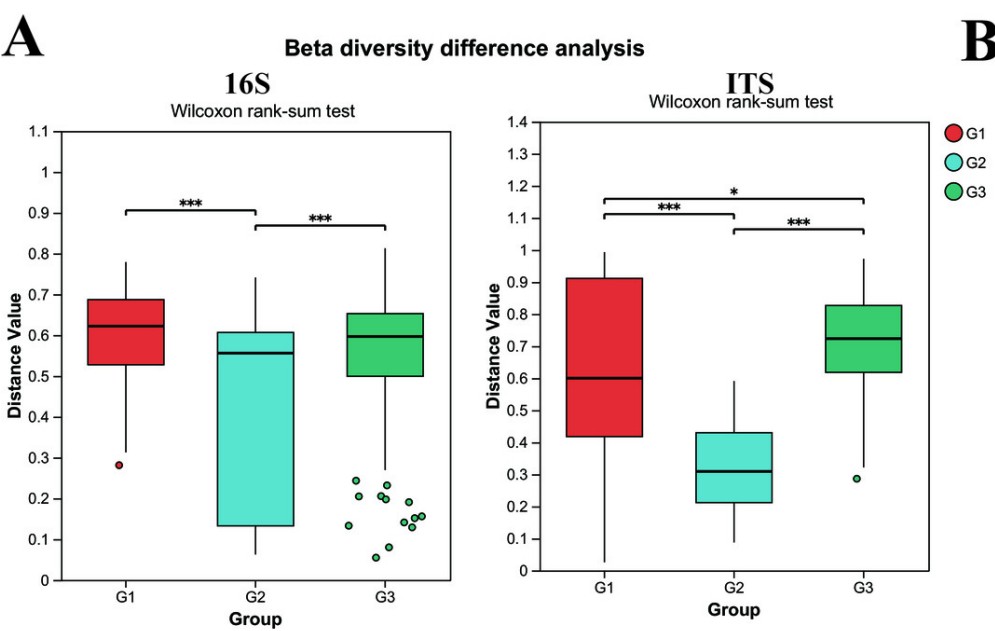

**Figure 3** **The differences of beta diversity between each group.** (A) Bacteria (16s) and (B) fungi (ITS) G1 represents neurotypical children; G2 represents severe ASD children; G3 represents mild ASD children.

suggest that the gut microbial beta diversity of children with ASD might be related to the severity of their symptoms.

Furthermore, we employed NMDS and PCoA to dissect the differences in beta diversity of gut bacteria and fungi among the three groups of children (Fig. 4). Firstly, both the NMDS and PCoA analyses demonstrated that it was challenging to distinctly discriminate the gut bacteria (16S rRNA) of the three groups of children on the coordinate axes. However, relatively significant differences still prevailed ($P < 0.05$). Meanwhile, in contrast to group G2, more samples from group G3 and group G1 (the neurotypical group) clustered within the same region. Unlike bacteria, the fungal communities exhibited conspicuous differences among the three groups of children, and these distinctions could be visualized on the coordinate axes. The gut fungal communities from children of different groups were evidently clustered within their respective regions. It is noteworthy that the results of the PCoA analysis indicated that the gut fungi of children in group G2 were markedly different from those in the other two groups, implying that the gut microbiota of children with severe ASD symptoms might have experienced more profound changes in comparison to the neurotypical group.

## Differences in gut bacterial and fungal communities between ASD and neurotypical children

The compositions of both bacteria and fungi were analyzed, ranging from the phylum level (Fig. S2) down to the genus level (Fig. 5), and then visualized in the form of a bar graph. In terms of bacteria, the five principal phyla present in all groups were Firmicutes, Bacteroidota, Proteobacteria, and Actinobacteria. However, the distribution patterns of the

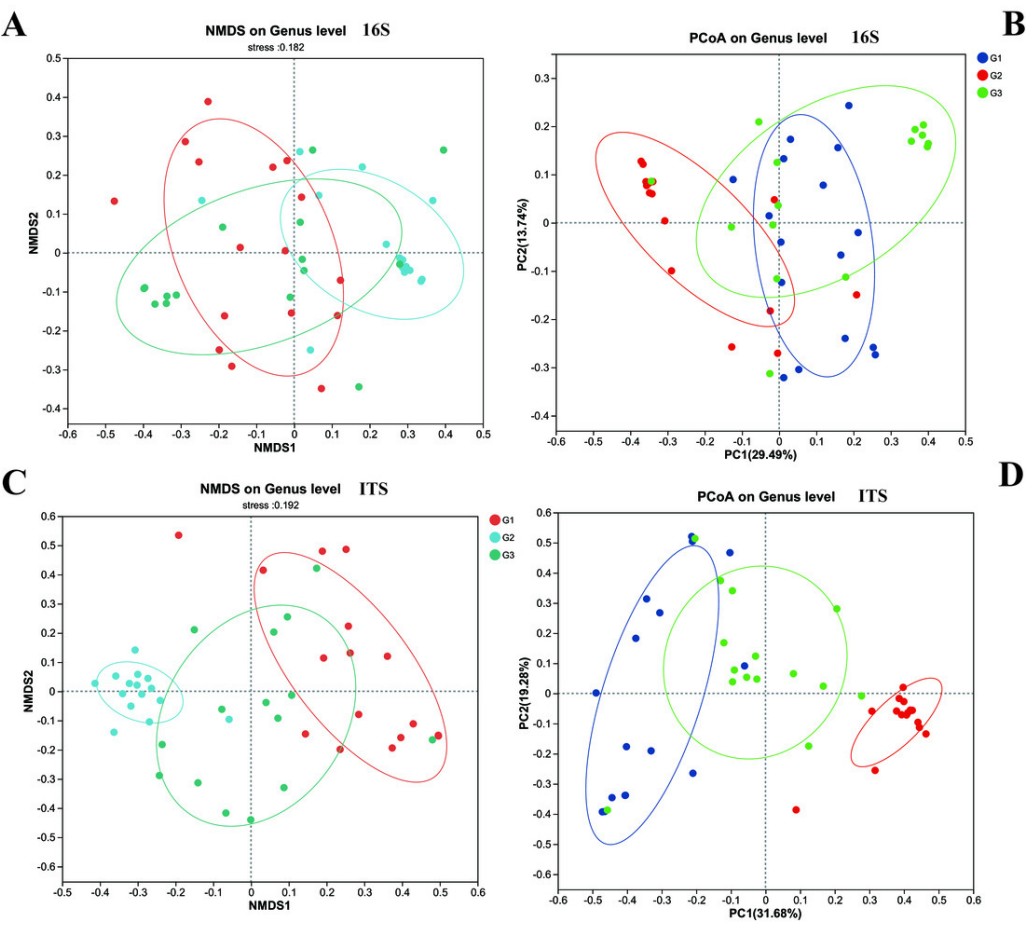

**Figure 4** **The beta diversity of three groups.** (A) 16s and (C) ITS NMDS analysis; (B) 16s and (D) ITS PCoA analysis.

abundances of these phyla differed between children with ASD and neurotypical children. Moreover, significant differences were detected when comparing children with severe ASD to those with mild ASD showed in the pie graph (Fig. S3). Among the neurotypical children, Firmicutes (63.30%) constituted the predominant phylum, trailed by Proteobacteria (17.63%), Bacteroidota (9.13%), and Actinobacteria (9.06%). Similarly, in children with severe ASD, Firmicutes (79.86%) was also the leading phylum, yet its relative abundance was elevated in comparison to that of neurotypical children. Concurrently, the relative abundance of Proteobacteria (1.40%) was markedly reduced. Likewise, children with mild ASD also exhibited a relatively high abundance of Firmicutes (56.56%), followed by Actinobacteria (18.46%), Proteobacteria (13.06%), and Bacteroidota (10.87%). These findings suggest that, at the phylum level, group G3 was more akin to group G1 in terms of community composition.

Similarly, when examining the phylum level of fungi, Ascomycota, Basidiomycota, and unclassified_k_fungi were identified as the three main fungal groups in all samples. In group G1, Ascomycota accounted for 65.78%, trailed by unclassified_k_fungi (26.32%)

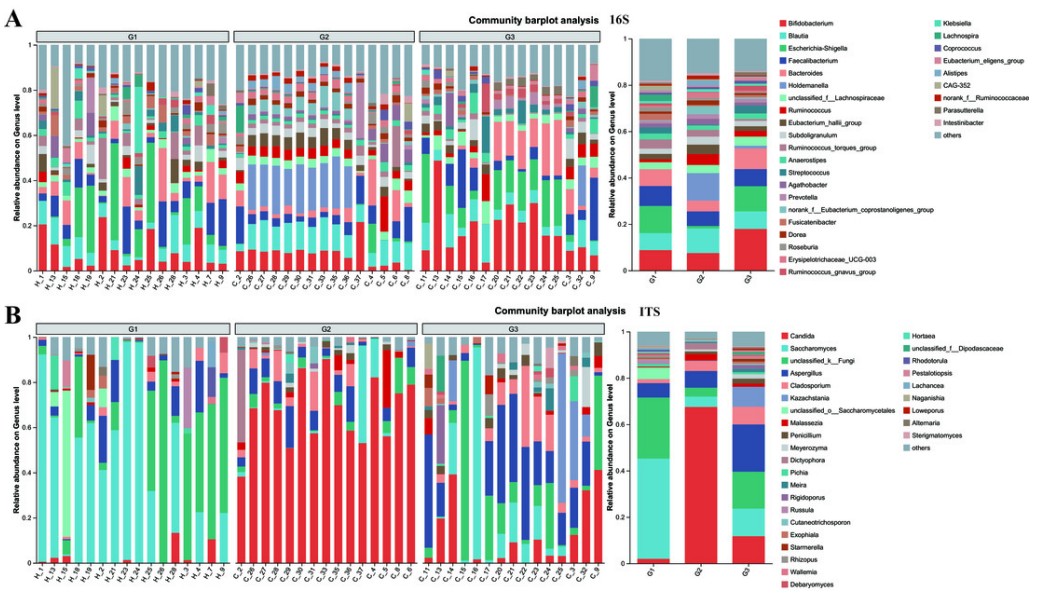

**Figure 5  The relative abundance of bacterial communities at genus levels.** (A) Bacteria of all samples; (B) fungi of all samples. G1 represents neurotypical children; G2 represents severe ASD children; G3 represents mild ASD children.

and Basidiomycota (6.30%). In group G2, the proportion of Ascomycota increased to 87.59%, accompanied by a slight increment in Basidiomycota (8.27%), while the unclassified_k_fungi exhibited a significant decrease (3.84%). Notably, when compared to group G2, the relative abundance of Ascomycota in group G3 decreased to 73.18%, and that of unclassified_k_fungi increased to 15.92%. These results imply that, in terms of fungal composition at the phylum level, group G3 was more similar to group G1 than to group G2.

At the genus level, notable differences emerge in the gut microbiota of the three groups of children with varying severities of ASD (Fig. 5 and Fig. S4). Firstly, the outcomes of 16S rRNA gene sequencing analysis for bacteria reveal that among neurotypical children, the top five most prevalent genera within the microbiome are *Escherichia-Shigella* (accounting for 11.7%), *Bifidobacterium* (8.8%), *Faecalibacterium* (8.61%), *Blautia* (7.33%), and *Bacteroides* (7.26%). In children with severe ASD, the predominant microbial genera are *Holdemanella* (11.88%), *Blautia* (10.6%), *Bifidobacterium* (7.53%), and *Faecalibacterium* (6.29%). For children with mild ASD, the leading microbial genera comprise *Bifidobacterium* (17.91%), *Escherichia-Shigella* (10.86%), *Bacteroides* (8.88%), *Blautia* (7.58%), and *Faecalibacterium* (7.40%).

The results obtained from the ITS analysis of fungi indicate that among neurotypical children, the principal genera consist of *Saccharomyces* (accounting for 43.19%), *Unclassified_k_Fungi* (26.36%), *Aspergillus* (6.23%), *Unclassified_o_Saccharomyces* (4.97%), and a minor proportion of Candida (2.01%). In children with severe ASD, *Candida* exhibits a substantial increase, reaching up to 67.52%, whereas the relative

abundance of *Saccharomyces* declines significantly (from 43.19% in neurotypical controls to 4.45% in severe ASD patients). Additionally, the fungal composition in these children also encompasses *Aspergillus* (7.22%), *Cladosporium* (4.22%), and *Unclassified_k_Fungi* (3.84%). Similarly, when compared with children having severe ASD, the main fungal constituents in children with mild ASD display a certain degree of restoration. This is manifested by the fact that the quantity of *Candida* is markedly lower than that in children with severe ASD, and the amount of *Saccharomyces* is greater than that in children with severe ASD. Meanwhile, regardless of whether in neurotypical children or those with mild ASD, the abundance of *Unclassified_k_Fungi* in the gut surpasses that in children with severe ASD. Consequently, from the vantage point of microbial composition, it appears that the principal gut microbiota of children with mild ASD bears a closer resemblance to that of neurotypical children, while the main gut microbiota of children with severe ASD diverges considerably from that of neurotypical children. This further suggests a potential association between the gut microbiota of children and relevant ASD symptoms.

To gain a more profound understanding of the situation, we decided to carry out a hierarchical clustering analysis on the top 50 principal microbial genera of both bacteria and fungi, which led to the generation of a species heatmap (Fig. 6). The heatmap indicates that, regardless of whether we are looking at bacteria or fungi, the gut microbiota of the three groups of children can generally be grouped together, with the clustering of fungi being more conspicuous. Interestingly, in terms of bacteria, group G1 appears to be relatively scattered, while groups G2 and G3 are relatively concentrated. It is rather difficult to determine whether group G1 is nearer to G2 or G3. However, upon conducting a detailed analysis of the ITS results, we would notice that the samples of G2 and G3 essentially form their own separate clusters. Moreover, some of the G1 samples are relatively close to the G3 samples and thus fall into the G3 cluster, such as H18, H4, H25, and so on. Notably, none of the G1 samples are included in the G2 cluster. These findings, taken together, suggest that the main composition of the microbiota in group G2 differs considerably from that of the gut microbiota of neurotypical children, and this difference might potentially lessen as the symptoms improve.

## Analysis of functional in gut bacteria and fungi between ASD and neurotypical children

Subsequently, the normal biological functions of bacteria and fungi in both neurotypical individuals and those with ASD were analyzed using PICRUSt2 and FUNGuild, with a one-way ANOVA test being employed. To further explore the differences among each group, the Tukey–Kramer test was utilized. The 16S rRNA functional analysis of bacteria unveiled that a total of 263 KEGG Level 3 modules were present across all samples. At Kyoto Encyclopedia of Genes and Genomes (KEGG) level 1, no remarkable differences were spotted among the three groups. However, when closely examining KEGG level 2 (Fig. 7A), it was found that significant differences existed between Group G2 and both Group G3 and Group G1 (with $P < 0.05$). In contrast, the difference between Group G1 and Group G3 was not statistically significant (as $P > 0.05$). Specifically, the amino acid metabolism in G2 ($P < 0.001$) was substantially higher than that in G1 and G3, while no

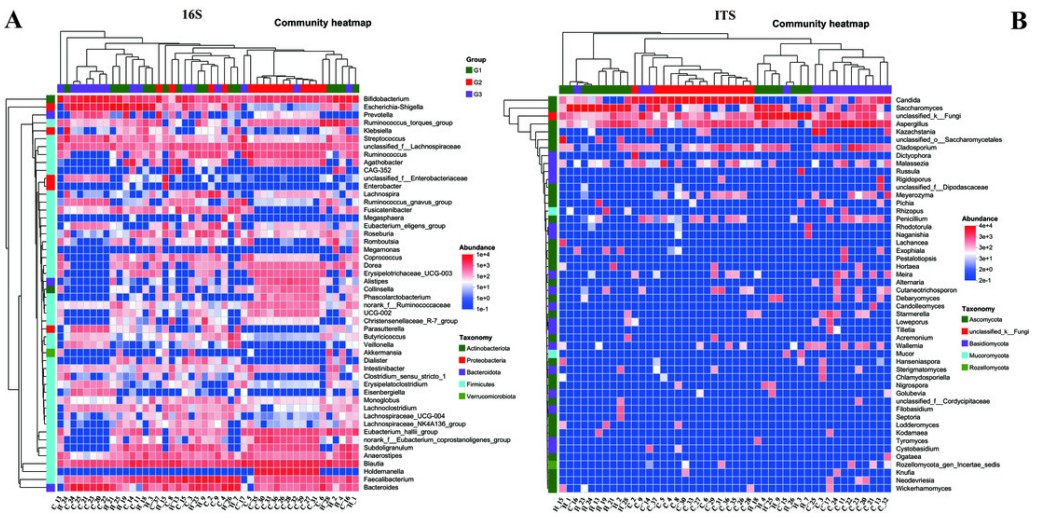

**Figure 6  Heatmap of the top 50 genera in all samples.** (A) Bacteria of all samples; (B) fungi of all samples.

difference was detected between G1 and G3 in this regard. Conversely, the Metabolism of cofactors and vitamins in both G2 and G3 ($P < 0.001$) was markedly lower than that in G1.

The functional prediction analysis carried out by FUNGuild demonstrated that the taxonomic variations of fungi in the intestines of children changed in line with different ASD symptoms (Fig. 7B). It also indicated that significant differences were present between G2 and G1 ($P < 0.05$ for Global and overview maps), whereas no significant difference was noted between G1 and G3. Among them, children with severe ASD had a higher proportion of undefined saprotroph ($P < 0.05$). Although other fungal taxonomic differences were not statistically prominent, a tendency could still be observed, suggesting that G1 was closer to G3. Collectively, these predictive analyses of bacterial functions and fungal functions provided solid evidence that the intestinal microbiota of children with severe ASD symptoms deviated from that of neurotypical children, while the intestinal microbiota of children with mild ASD seemed to bear a closer resemblance to that of neurotypical children. Hence, it can be concluded that the composition and functions of intestinal bacteria and fungi exerted a certain degree of influence on ASD symptoms.

## DISCUSSION

Here, we focus on the differences in gut microbiota between children with different ASD symptoms and neurotypical ones, and explain these differences from aspects of diversity, composition, and functional prediction. The aim is to deeply understand the role of gut microbiota in ASD pathogenesis, the relationship between ASD symptoms and gut microbiota, and lay a theoretical foundation for ASD diagnosis and treatment strategies based on microbiota.

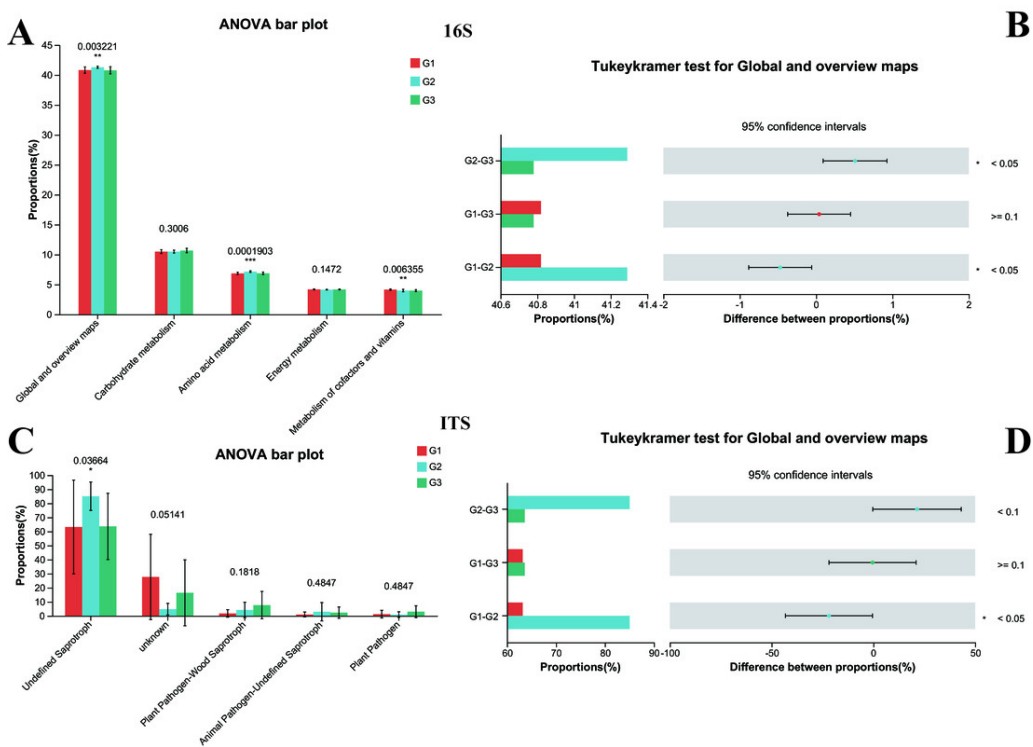

**Figure 7** **Functional analysis of three groups.** The significance differences between each group were indicated with an asterisk (*), and $P < 0.05$ was listed. (A) 16s proportions of different bacteria functional at level 2 by KEGG. (B) 16s differences between each group. (C) ITS proportions of different fungal functional. (D) ITS differences between each group.

In terms of gut microbial diversity, a plethora of studies have unequivocally illustrated significant discrepancies between children with ASD and neurotypical children (*Xie et al., 2022*; *Zarimeidani et al., 2024*). In the current study, the α diversity analysis revealed that no conspicuous differences were detected between severely affected ASD children and the neurotypical group. However, notable differences emerged when comparing severely affected ASD children with mildly affected ASD children. This implies that the gut bacterial diversity exhibits distinct characteristics in children with varying degrees of ASD. To date, the differences in α diversity between ASD children and the neurotypical group remain inconsistent. For instance, *Son et al. (2015)* reported no observable alterations in the diversity and richness of the gut microbiota in the stools of ASD subjects and their neurotypical sibling controls. Conversely, *De Angelis et al. (2015)*, *Finegold et al. (2010)* discovered greater microbial diversity in ASD subjects than in controls. Consequently, gut microbiota dysbiosis serves as a more accurate indicator for elucidating the changes in microbial community structure. In the examination of the GMHI and the MDI, a significant shift was observed in the GMHI of children with ASD. Notably, the degree of microbiota dysbiosis was found to be closely intertwined with the severity of ASD symptoms. Children with milder symptoms exhibited a gut microbiota health status that was more proximate to

that of neurotypical children, suggesting a potential relationship between ASD symptoms and gut microbiota diversity.

$\beta$ diversity analysis provides further robust support for this perspective. Children with severe ASD display significant disparities in both bacterial and fungal $\beta$ diversity when compared to both the neurotypical group and those with mild ASD. In contrast, the differences between the neurotypical group and children with mild ASD are comparatively minor. This unambiguously demonstrates that the diversity of the gut microbial community structure is intimately correlated with the severity of ASD symptoms, and that the gut microbial community structure of children with milder symptoms bears a closer resemblance to that of neurotypical children. Up to now, numerous studies have illustrated alterations in the microbial community structure within the ASD group. However, only a limited number of studies have concentrated on the differential severities of ASD in children. In our current study, we not only corroborated the existence of significant differences in the gut microbial community structure between children with ASD and neurotypical children but also elucidated that such differences were linked to the disease severity.

Regarding the composition of gut microorganisms, both bacteria and fungi manifest significant discrepancies between children with ASD and neurotypical children at the phylum and genus levels. In the context of bacterial composition, Firmicutes, Bacteroidetes, Proteobacteria, and Actinobacteria constitute the principal phyla in all groups. However, their abundance distributions vary considerably between children with ASD and their neurotypical peers, and substantial differences also exist between severely affected ASD children and those with milder symptoms. Previous investigations have already attested to the alterations in the relative abundances of Firmicutes, Bacteroidetes, Proteobacteria, and Actinobacteria when comparing ASD and neurotypical children. Concurrently, certain researchers have reported a trend towards a diminished proportion of *Bacteroidetes* and an elevated level of Firmicutes in children with ASD (*Strati et al., 2017*; *Tomova et al., 2015*; *Williams et al., 2011*). In our present study, Firmicutes accounted for the highest proportion among neurotypical children, whereas in ASD children, the relative abundance of Firmicutes was found to increase even further, corroborating the earlier findings. When comparing the phylum taxa of fungi, a significantly higher proportion of Ascomycota was observed in children with severe ASD. In contrast, no significant disparity was detected between children with mild ASD and neurotypical children. The elevation of Ascomycota in children with ASD has been corroborated by previous studies (*Retuerto et al., 2024*). Thus, our study suggests that the relative abundance of the Ascomycota phylum could potentially serve as an indicator for ASD symptoms.

At the genus level, the dominant genera of gut microorganisms differ among neurotypical children, those with severe ASD, and those with mild ASD. Notably, the relative abundance of *Holdemanella* was markedly elevated, whereas the relative abundances of *Bifidobacterium* and *Faecalibacterium* exhibited a slight decline. Although only a few studies have reported a higher relative abundance of *Holdemanella* in children with ASD, a significant increase in *Holdemanella* has been observed in adults with ASD (*Zhang et al., 2021*). *Bifidobacterium* and *Faecalibacterium* are both beneficial bacteria, and previous research has detected

their decreased presence in children with ASD (*Larroya-Garcia, Navas-Carrillo & Orenes-Pinero, 2019*; *Retuerto et al., 2024*). It is worth highlighting that the bacterial composition in children with mild ASD is more akin to that of neurotypical children. In terms of fungi, at the genus level, yeasts and unclassified fungi are the predominant genera in neurotypical children. In children with severe ASD, *Candida* shows a significant upsurge, while *Saccharomyces* experiences a sharp decline. The genus *Candida*, belonging to the Ascomycota phylum, has been extensively demonstrated to be more abundant in children with ASD than in neurotypical children (*Iovene et al., 2017*; *Retuerto et al., 2024*; *Strati et al., 2017*). Previous reports have proposed that the growth of *Candida* in the intestines might lead to reduced absorption of carbohydrates and minerals, as well as elevated toxin levels, which are hypothesized to contribute to autistic behaviors (*Kantarcioglu, Kiraz & Aydin, 2016*). Conversely, *Saccharomyces* is commonly regarded as a probiotic and has been shown to be negatively correlated with the scores of ASD core symptoms (*Li et al., 2024*; *Raghavan et al., 2023*). Meanwhile, in this study, the fungal composition in children with mild ASD also demonstrates a tendency to revert towards that of neurotypical children. These findings strongly suggest that alterations in the gut microbiota of children are associated with ASD symptoms.

In terms of the functional prediction analysis of bacteria and fungi, substantial differences have emerged among different groups. By employing PICRUSt2 to analyze bacterial functions, significant disparities were detected at KEGG level 2 between children with severe ASD, the neurotypical group, and those with mild ASD. For instance, the amino acid metabolism in children with severe ASD was considerably higher than that in the neurotypical group and those with mild ASD, whereas the metabolism of cofactors and vitamins was significantly lower than that of the neurotypical group. The study conducted by *Jones et al. (2022)* also identified differences in the metabolic functions of gut microorganisms between children with ASD and neurotypical children, further corroborating the findings of this study. When it came to analyzing fungal functions using FUNGuild, significant differences were observed between children with severe ASD and the neurotypical group in the global and overview maps. Notably, the proportion of undefined saprotrophs was higher in children with severe ASD. These results of functional prediction analysis comprehensively indicate that the functions of gut microorganisms have undergone alterations in children with ASD, and the extent of deviation of children with severe ASD from neurotypical children is more pronounced. In contrast, the gut microbial functions of children with mild ASD appear to be closer to those of neurotypical children. This further underlines the significance of gut microbial functions in the pathological process of ASD and the intimate relationship between the gut microbiota and ASD symptoms, thereby providing potential targets and directions for ameliorating ASD symptoms by modulating gut microbial functions in the future.

In this study, we investigated the association between ASD and the gut microbiota, demonstrating a close relationship between alterations in the gut microbiome and ASD symptoms. However, due to limitations in sample collection and analytical costs, we did not perform metabolomic profiling of gut microbial metabolites, nor did we establish a direct link between ASD symptoms and microbial metabolic products. Notably, our study

lacks an in-depth discussion of the gut-brain axis through metabolite analysis. Therefore, in future research, we aim to explore the association between specific ASD symptoms and gut microbial metabolism, which may provide a foundation for developing microbiota-targeted therapeutic interventions for ASD.

## CONCLUSION

In summary, this study has disclosed the disparities in the gut microbiota between autistic children exhibiting varying symptoms and their neurotypical counterparts. It has elucidated the relationship between the gut microbiota and ASD symptoms. Notably, it has been demonstrated that children with milder ASD symptoms bear a closer resemblance to neurotypical children in terms of gut microbial diversity, composition, and function. This not only furnishes solid evidence but also presents novel ideas for further probing into the pathogenesis of ASD and for devising precise intervention strategies predicated on the gut microbiota.

## ACKNOWLEDGEMENTS

We thank the employees of Affiliated Hospital of Putian University for their help with sample collections. We also thank the reviewers and editors for the constructive comments to improve our manuscript.

### Funding

This study was funded by grants of the National Natural Science Foundation of China (82302565), the Natural Science Foundation of Fujian Province (2024J011459), Putian Science and Technology Bureau (2022SZ3001ptxy08), Startup Fund for Advanced Talents of Putian University (2021068, 2021069). The funders had no role in study design, data collection and analysis, decision to publish, or preparation of the manuscript.

### Grant Disclosures

The following grant information was disclosed by the authors:
National Natural Science Foundation of China: 82302565.
Natural Science Foundation of Fujian Province: 2024J011459.
Putian Science and Technology Bureau: 2022SZ3001ptxy08.
Startup Fund for Advanced Talents of Putian University: 2021068, 2021069.

### Competing Interests

The authors declare there are no competing interests.

### Author Contributions

- Qinghuang Zeng performed the experiments, analyzed the data, prepared figures and/or tables, authored or reviewed drafts of the article, and approved the final draft.

- Yisheng Hu performed the experiments, analyzed the data, prepared figures and/or tables, and approved the final draft.
- Leiying Xie performed the experiments, analyzed the data, prepared figures and/or tables, and approved the final draft.
- Xinyi Zhang performed the experiments, analyzed the data, prepared figures and/or tables, and approved the final draft.
- Yun Huang performed the experiments, prepared figures and/or tables, and approved the final draft.
- Jianbin Ye conceived and designed the experiments, analyzed the data, authored or reviewed drafts of the article, and approved the final draft.
- Shouan Wang conceived and designed the experiments, prepared figures and/or tables, and approved the final draft.
- Jia Xu conceived and designed the experiments, authored or reviewed drafts of the article, and approved the final draft.

### Human Ethics

The following information was supplied relating to ethical approvals (i.e., approving body and any reference numbers):

The Ethics Committee of the Affiliated Hospital of Putian University granted Ethical approval to carry our the study within its facilities (Approval No.: 2023090-1).

### Data Availability

The raw data is available at NCBI SRA: PRJNA1207206.

The raw sequence data are available at FigShare: Ye, Jianbin (2025). Gut microbiota diversity and composition in children with autism spectrum disorder: associations with symptom severity. figshare. Dataset. https://doi.org/10.6084/m9.figshare.28303508.v1.

### Supplemental Information

Supplemental information for this article can be found online at http://dx.doi.org/10.7717/peerj.19528#supplemental-information.

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
