# Peer review of "Gut microbiota diversity and composition in children with autism spectrum disorder: associations with symptom severity"

_PeerJ, doi:10.7717/peerj.19528_

## Round 0.1 · original submission · Major Revisions

Dear Dr. Ye, I hope that you will improve the manuscript very carefully according to the reviewers' comments. I hope that this will enable the reviewers to approve the new version of this manuscript.

Reviewer 1 ·

Basic reporting

Overall, this manuscript requires tons of improvements. The authors need to clearly explain the rationale behind this experiment: What is the relationship and distinction between your samples or experimental design and those in previous studies? What is the scientific hypothesis you aim to test? How were the samples processed and how was the data handled? In accordance with the scientific hypothesis, how should the results be interpreted, particularly the differences between severe and mild ASD patients? Also, please be very cautious in selecting references and avoid overinterpreting the results of the cited papers.

Experimental design

The experiment was poorly designed. The Introduction lacks a clear organization of the research background and an accurate summary of the study’s objectives. Although it lists numerous published studies related to ASD, it didn’t provide rationally logical connection between these examples. For example, which studies are based on adult cases and which on pediatric cases? Since several studies have already reported the gut microbiota characteristics in ASD children and identified potential biomarkers, what distinguishes this study from previous work? Are there any particular aspects of the experimental design or samples? What is the novelty of this study? What is the specific scientific hypothesis being tested?

Materials and Methods section does not clearly describe the sample processing procedures. In fact, the authors seem unclear about which data processing method was used. For example, lines 197–200 state that “sequences... were denoised using either the DADA2 plugin or the Deblur plugin within Qiime2”—which one was actually used??? Similarly, lines 200–202 mention that “sequences annotated as chloroplast and mitochondrial in all samples were removed (in case of chloroplast and mitochondrial contamination in the samples, removal was recommended)”—who recommended this removal??? The same issue arises in lines 203–204 with the phrase “was rarified to 20,000 (a recommended rarefaction approach).” Furthermore, lines 206–207 state that “ASVs were taxonomically classified using either the Naive Bayes, Vsearch, or Blast classifier in Qiime2.” Again, which method was actually used???? Additionally, while lines 212–213 claim that “the Alpha diversity indices Ace were computed,” Figure S1 in the results shows plots based on the Shannon index. Finally, did the authors use all ASVs output by DADA2 or another software for data analysis, and were low-frequency ASVs removed? If so, what was the threshold? There are too many details that require clarification.

Validity of the findings

The quality of the Discussion section is very poor. It lacks proper interpretation of the data. The authors listed numerous related studies without offering a clear viewpoint. This might stem from an unclear research objective. As noted in the evaluation of the Introduction section, the authors need to clarify how this study differs from previous ones, what is unique about the experimental design or samples, and what specific scientific hypothesis is being tested. Only then can they logically restructure the discussion and provide a detailed interpretation and speculation—perhaps by interpreting the results according to patient age, microbial differences, and the functions influenced by the microbiota. Moreover, I checked some of the references cited by the authors (e.g., line 450: “Peng et al. 2009; Wang et al. 2012”), and I do not believe that these papers discuss issues related to propionate metabolism or the intestinal immune barrier. The authors must have great care in interpreting their data and in citing literature, avoiding overinterpretation.

Additional comments

There are many minor typos throughout the text and figures—for example, the inconsistent use of “16s” and “16S.” Also, only Latin names at the genus level should be italicized; names at other taxonomic levels and other parts of the English text do not require italics. Some figures, such as Fig. 6, are of poor quality and it is difficult to see some of the text. The details of the English language should be corrected by a native speaker.

·

Basic reporting

Dear Authors,

The manuscript discusses the association of gut microbiome composition with symptoms in ASD. The scientific content of the study will contribute to the research community, as the investigation of individual factors that illuminate aspects of the disease is crucial for creating optimal therapeutic approaches.
However, some modifications are necessary.

Experimental design

The introduction establishes a connection to previous research and outlines the research objective. However, it should briefly mention the research gap that motivates the study and its findings. Additionally, the introduction could benefit from including more studies on the interaction between the gastrointestinal system and the brain, particularly regarding the gut-brain axis and its impact on the cognitive and social development of individuals with ASD.

I suggest the following as an example:

https://doi.org/10.3390/applmicrobiol4040114
https://doi.org/10.3390/ijms19082251
https://doi.org/10.1038/s41593-023-01361-0

Validity of the findings

Furthermore, it would be advantageous to include a separate paragraph in the discussion addressing the study's limitations and suggestions for future research. Expanding the conclusions would also enhance the overall quality of the manuscript.

---

## Round 0.2 · accepted · Accept

Dear Dr. Ye,

I am pleased to inform you that this article has been accepted for publication. I hope you will continue further research in this very important direction.

·

Basic reporting

The authors have made significant revisions to the manuscript, addressing all the suggested modifications. As a result, the quality of the content has improved.

Experimental design

No comment.

Validity of the findings

No comment.

Reviewer 3 ·

Basic reporting

no comment

Experimental design

The manuscript was written with high quality, the authors carefully planned the experiments, processed the data efficiently, and optimally displayed them in the Results section.

Validity of the findings

Conclusions are well stated, linked to original research question & limited to supporting results.

Additional comments

The manuscript is well written, the authors conducted a fairly large and high-quality study worthy of publication. The results will be of interest not only to practicing doctors, but also to specialists in the field of microbiology, biochemistry, and physiology. The authors have presented the results of the research quite well.